plant science/genetics/evolution

domestication, gene flow, introgression, maize, Mexico, teosinte

**Authors for correspondence:**
Alejandra Moreno-Letelier
e-mail: amletelier@ib.unam.mx
Luis E. Eguiarte
e-mail: fruns@unam.mx

# The relevance of gene flow with wild relatives in understanding the domestication process

Alejandra Moreno-Letelier[1], Jonás A. Aguirre-Liguori[2], Daniel Piñero[2], Alejandra Vázquez-Lobo[3] and Luis E. Eguiarte[2]

[1]Jardín Botánico, Instituto de Biología, Universidad Nacional Autónoma de México, Ciudad de México, México
[2]Departamento de Ecología Evolutiva, Instituto de Ecología, Universidad Nacional Autónoma de México, Ciudad de México, México
[3]Centro de Investigación en Biodiversidad y Conservación, Universidad Autónoma del Estado de Morelos. Av. Universidad 1001 Cuernavaca, Morelos, 62209, México

AM-L, 0000-0001-7524-7639; JAA-L, 0000-0003-1763-044X; AV-L, 0000-0002-7828-1653; LEE, 0000-0002-5906-9737

The widespread use of genomic tools has allowed for a deeper understanding of the genetics and the evolutionary dynamics of domestication. Recent studies have suggested that multiple domestications and introgression are more common than previously thought. However, the ability to correctly infer the many aspects of domestication process depends on having an adequate representation of wild relatives. Cultivated maize (*Zea mays* ssp. *mays*) is one of the most important crops in the world, with a long and a relatively well-documented history of domestication. The current consensus points towards a single domestication event from teosinte *Zea mays* ssp. *parviglumis* from the Balsas Basin in Southwestern Mexico. However, the underlying diversity of teosintes from *Z. mays* ssp. *parviglumis* and *Zea mays* ssp. *mexicana* was not taken into account in early studies. We used 32 739 single nucleotide polymorphisms (SNPs) obtained from 29 teosinte populations and 43 maize landraces to explore the relationship between wild and cultivated members of *Zea*. We then inferred the levels of gene flow among teosinte populations and maize, the degree of population structure of *Zea mays* subspecies, and the potential domestication location of maize. We confirmed a strong geographic structure within *Z. mays* ssp. *parviglumis* and documented multiple gene flow events with other members of the genus, including an event between *Z. mays* ssp. *mexicana* and maize. Our results suggest that the likely ancestor of maize may have been domesticated in Jalisco or in the southern Pacific Coast

and not in the Balsas Basin as previously thought. In this context, different populations of both teosinte subspecies have contributed to modern maize's gene pool. Our results point towards a long period of domestication marked by gene flow with wild relatives, confirming domestication as long and ongoing process.

# 1. Introduction

Recent genomic studies have revealed that domestication is a complex process often defined by multiple origin events and recurrent gene flow between cultivars and their wild relatives. For example, multiple domestication events with gene flow have been inferred for pigs, rice, barley, chili pepper and common bean [1–4]. As a consequence, the protracted model of domestication [5]—which proposes that crops were domesticated over a long time period punctuated by gene flow events—has gained wide acceptance.

By contrast, the current paradigm of maize domestication points to a single domestication event (figure 1a) [6,7]. However, recent studies have suggested a more complex scenario with multiple events of gene flow and selection (figure 1b,c), which would be consistent with the high mobility of ancient Mesoamerican peoples and adaptation to different environments [7–13].

The *Zea mays* species complex consists of four subspecies: the domestic maize (*Zea mays* ssp. *mays* (Schrad. Iltis) and three wild relatives collectively called teosinte. One of the wild relatives, *Zea mays* ssp. *mexicana* (hereafter *mexicana*), grows predominantly at high elevations (1600–2700 m) in the relatively dry regions of Central Mexico. Another wild relative, *Zea mays* ssp. *parviglumis* (hereafter *parviglumis*; [14]) is adapted to the warmer, low to middle elevations of Southwestern Mexico (less than 1800 m) [15]. On the basis of morphological data, a fourth subspecies has also been identified, *Zea mays* ssp. *huehuetenangensis* Iltis & Doebley, found in Guatemala [14]. Genetically, some *parviglumis* populations are highly differentiated, particularly those from some regions of the states of Jalisco and Guerrero [8,16–18].

The current paradigm for the domestication of maize points to a single origin from *parviglumis* somewhere in the Balsas Basin of Southwestern Mexico (figure 2, populations Teloloapan (1), Alcholoa (2) and Huitzuco (8) [9,19,20]. Some archaeological evidence is consistent with this lowland domestication scenario, given excavations of the central Balsas valley that have unearthed the earliest (8700 years BP) phytolith evidence for maize cultivation [21], as well as genetic evidence from isozyme and microsatellite data [19,22]. However, a Balsas domestication is not completely consistent with single nucleotide polymorphism (SNP) data [20].

There are at least two factors that have complicated inferences about the domestication and evolution of maize. Multiple lines of evidence indicate that the adaptation of maize from the lowlands to the highlands was aided by introgression from *mexicana*; this introgression probably occurred in the past [23,24] and may still be ongoing [24–26]. Hence, while maize may have been domesticated from *parviglumis*, its genomic composition is also a product of *mexicana*, particularly in maize growing in the Mexican highlands [24]. The introgression has probably obscured some of the historical features of the initial domestication of maize [20], including the observation that maize from the highlands (and not the lowland Balsas region) are apparently genetically closer to *parviglumis* in some analyses [19,20]. In short, a history of introgression between maize and teosintes has influenced the genetic composition of extant maize and, therefore, obscured its domestication origin [23,27].

A second complication is the location of maize domestication. The foundational isozyme study that established *parviglumis* as the ancestor of maize was unable to differentiate whether *parviglumis* from the Balsas or Jalisco populations were more closely related to the Mexican landraces of maize [22]. Subsequent studies tentatively concluded that maize was more closely related to teosinte from the Balsas region [28]. It is worth noting, however, that the Jalisco region was not well represented in these studies, since 84% of the *parviglumis* samples were from the Balsas populations [22]. Similarly, the landmark microsatellite study that supported a single maize domestication event from *parviglumis* included 34 plants from throughout the geographic distribution of *parviglumis*, but only five plants (20%) were from Jalisco, one from Nayarit, another from Colima and the rest mainly from the Balsas region [19]. The limited sampling of the highly diverse Jalisco region means that previous analyses may not have included the full extent of *parviglumis* diversity and therefore may have had limited power to identify the most likely origin of domesticated maize.

More recent genome-based analyses provide hints that the Balsas origin may not be correct. For example, a SNP-based study investigated the origin of maize by inferring maize ancestral allele frequencies and comparing them with those of extant teosinte accessions, showing that maize landraces from the Jalisco region have allele frequencies that were closest to the inferred ancestral

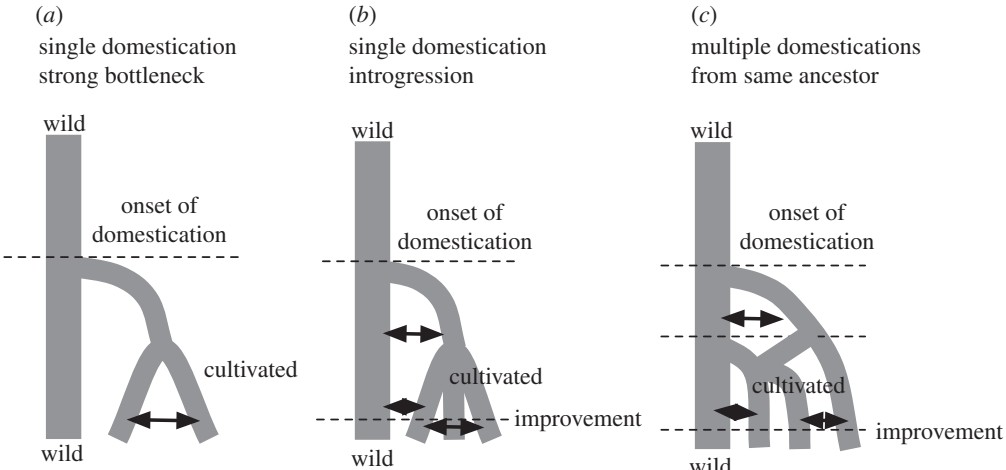

**Figure 1.** Demographic domestication models involving a single domestication event (*a*), a single domestication with ongoing gene flow (*b*) and multiple domestication/improvement events with gene flow (*c*). Modified from [6].

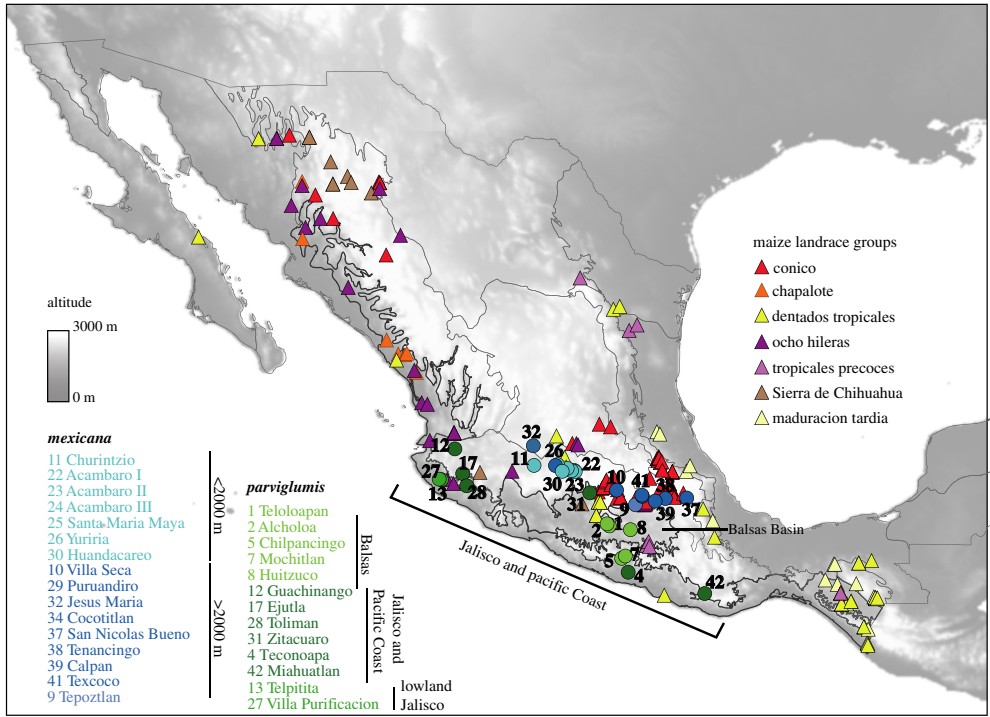

**Figure 2.** Distribution of sampling sites of analysed teosinte varieties and maize landraces. The main biogeographic regions represented in this study are highlighted and indicated in the figure. Numbers indicate the different populations and colours correspond to those of figure 3*a*.

maize allelic frequencies [20]. This same study used a spatial approach to identify geographic regions where maize landraces have the least drift from inferred ancestral allele frequencies; these regions again included some of the higher altitude portions of the state of Jalisco.

A more recent analysis of population genomic data placed the time of divergence between maize and four Balsas accessions at approximately 75 000 years BP [29], which greatly exceeds the accepted domestication time of approximately 9000 years [30], but is closer to the 55 000 years BP estimated by nuclear sequence markers [10]. This discrepancy in divergence dates may suggest that the four Balsas accessions used for molecular dating do not represent lineages derived from the source populations for maize domestication.

Given the limitations of current analyses, we believe that the western distribution of *parviglumis* merits additional attention based on two additional lines of reasoning. The first is that parts of the

state of Jalisco, like the Balsas Basin, have been identified as a potential refugia during the warm period of the mid-Holocene [15]. Moreover, ecological modelling suggests that *parviglumis* was distributed across the Jalisco and Balsas regions dating back to the last glacial maximum, approximately 21 000 years BP [15]. The second is that archaeological evidence of Early-Holocene agriculture has been found in this region (see review in [31]), just as in the Balsas Basin [9]. Third, Jalisco state is the centre of domestication for two species of bean (*Phaseolus vulgaris* and possibly *P. acutifolius*) [2,32–34]. Thus, both climatological and cultural data suggest the possibility of maize domestication in the Jalisco-southern Pacific Coast region.

Historically, *parviglumis* from the Jalisco region has been understudied, but recent population genetic surveys have included more Jalisco populations. For example, Pyhäjärvi *et al.* [16] genotyped 72 individuals from six Jalisco populations with the MaizeSNP50 bead chip. Their results showed that *parviglumis* populations are genetically structured and that Jalisco populations generally contain high levels of genetic diversity. More recently, our group used the MaizeSNP50 bead chip to genotype 28 teosinte populations including a broad sampling of Jalisco [16–18]. These analyses confirmed high genetic diversity within *parviglumis* populations and a strong genetic differentiation among Jalisco populations [16–18].

In this study, we combined genotypic data from 29 teosinte populations, which represent the entire range of *parviglumis* and *mexicana* (with the exception of some northern populations, excluded because they are very isolated and atypical, see [17]) and include the six populations from Jalisco (50% of all *parviglumis* populations), five populations from the Balsas Basin and two from the Pacific Coast of Oaxaca and Guerrero, with 43 maize landraces (details in electronic supplementary material, table S2; [35,36]). Altogether, our genotype data include 403 individuals (116 from maize, 205 from *parviglumis* and 211 from *mexicana*) and 30 673 SNPs [37]. We used this data to explore relationships between maize, *mexicana, parviglumis* and other members of *Zea* and to investigate the history of introgression among the subspecies of *Zea mays* to propose an alternative domestication centre for maize.

# 2. Material and methods

## 2.1. Sampling

Plant material was collected from 28 populations of teosinte, 15 populations of *mexicana* and 13 populations of *parviglumis* for a total of 403 individuals, which covers the entire range of geographical and environmental conditions of both subspecies (figure 2). Details of the sampling design, DNA extraction, genotyping and locality information are described in [11,17], and in the electronic supplementary material.

Maize samples were obtained from 161 individuals of 46 landraces from the main landrace groups of Mexico, averaging 3.6 individuals per landrace. Information about sampling, DNA extraction and genotyping can be found in Arteaga *et al.* [35].

## 2.2. Single nucleotide polymorphism calling and data quality assessment

SNPs were detected with the Illumina Maize SNP50 Bead Chip as described for maize by Arteaga *et al.* [35] and for teosinte by Aguirre-Liguori *et al.* [17]. Automated allele calling was performed using GenomeStudio 2010.1 (Genotyping module 1.7.4; Illumina), excluding those loci with a GenTrain score less than 0.3, and those with more than 10% of missing data. Scores between 0.3 and 0.45 were manually checked and curated. After filtering, a total of 34 981 SNPs were recovered. To reduce the redundancy of the data, loci in strong linkage disequilibrium ($r^2 > 0.8$) were removed using Plink 1.07 [38], for a total of 32 739 loci. Additionally, information from the Illumina Maize SNP50 Bead Chip of *Zea luxurians* ($n = 6$), *Z. diploperennis* ($n = 6$) and *Z. perennis* ($n = 6$) were added to some of the analyses, in order to have a proper outgroup for the *Zea mays* complex. Data was obtained from Trtikova *et al.* [39]. The shared loci between our dataset and that of [39] yielded a total of 30 673 SNPs.

To evaluate the effect of ascertainment bias in our data, parallel analyses were performed using the genotyping-by-sequencing (GBS) teosinte and maize dataset from Swarts *et al.* [27], by filtering all sites with over 50% of missing data and those with missing allele frequencies under 0.05 using vcftools [40]. The dataset had 165 192 SNPs, but after removing sites with gaps, the total was 122 085 SNPs.

## 2.3. Genetic differentiation and genetic structure

Genetic differentiation was explored with a principal component analysis (PCA) with all the 30 673 SNPs of maize, teosinte and outgroups, using the package TASSEL 5 [41]. A genetic distance matrix was constructed using TASSEL 5 by estimating 1-IBS, where IBS is the probability that two alleles drawn at random from two individuals at the same locus are the same (identity by state) [41]. The advantage of using this metric of genetic distance for SNP data is that the genetic information in heterozygotes is not lost, as it happens with traditional Nei's distance methods, where ambiguity codes are treated as missing data. From the distance matrix, we constructed a rooted dendogram using *Z. luxurians*, *Z. perennis* and *Z. diploperennis* as outgroups [41]. The same analysis was performed with the GBS data [27], using a 1-IBS distance matrix and rooting with *Z. luxurians*, and the resulting relationships were displayed with a network with the NeighborNet algorithm implemented by SplitsTree [42].

Genetic clustering analyses were performed with fastStructure [43]. The number of *K* values tested ranged from 1 to 10, with 10 iterations each. The optimal *K* values were evaluated with the Evanno method [44].

## 2.4. Gene flow and ancestral introgression

In general, it is difficult to discern between shared ancestral polymorphism and gene flow [45]. To explore this, we used TreeMix, which is based on a maximum-likelihood population graph and identifies pairs of populations with a higher covariance of allelic frequencies than expected under the no migration model [46]. The ancestral allele frequencies are inferred using *Z. perennis* and *Z. diploperennis* as an outgroup [39]. Bootstrap values were obtained from 100 replicates using the -b option of TreeMix [46]. The consensus tree was obtained using the Majority Rule option in the package Consense implemented by WebPhylip v. 2.0 (http://bar.utoronto.ca/webphylip/) and visualized with FigTree v.1.3.1 (https://github.com/rambaut/figtree/releases). The analysis was carried out with the 30 673 SNPs dataset including the *Z. perennis* and *Z. diploperennis*.

To better represent gene flow between populations on a geographic scale, we performed a geographically explicit ancestry analysis using all teosinte populations, and 'conico' maize accessions which showed admixture in the fastStructure analysis and have been reported to have introgressed with *mexicana* [24,26]. Only the accessions from the highlands of central Mexico were included, because they are potentially sympatric with *mexicana* populations. We included all 403 georeferenced samples of teosinte (both *parviglumis* and *mexicana*) and 44 georeferenced maize accessions belonging to the following landraces of the conico group: conico (*n* = 30), arrocillo (*n* = 4), chalqueño (*n* = 5) and cacahuacintle (*n* = 5). The analysis was performed with TESS3 [47]. Unlike the fastStructure admixture analysis, no outgroups or lowland maize accessions were included.

# 3. Results and discussion

We first analysed genetic structure within our dataset with a PCA; this revealed three interesting features (figure 3a). The first was that there is a clear differentiation between maize landraces and the teosintes. These two groups were separated by the first principal component, which described 10% of the variation. This observation is initially surprising, since maize is derived from *parviglumis*, so one might expect the differentiation between maize and the teosintes would be lower. Nevertheless, this observation has been made previously with both ascertained (SNPchip; [37]) and non-ascertained (GBS; [27]) data. The strong PCA differentiation between maize and the teosintes can be attributed to accelerated genetic drift and/or selective sweeps during domestication [48]. In figure 3a, we also observe *mexicana* populations grouped into a single cluster. Finally, as previously observed [16–18], *parviglumis* populations were differentiated into more than one cluster (figure 3a), with the Jalisco populations from Telpitita (**13**) and Villa Purificacion (**27**) on the Pacific Coast forming a distinct cluster from the other Jalisco and Pacific Coast populations despite being geographically close (figure 3a).

We also constructed a dendogram of individuals, based on IBS over all loci. The network shows individuals strongly clustered by population and those branches were collapsed (electronic supplementary material, figure S1). Our results also place populations of *parviglumis* from Jalisco more closely to maize than most populations of *parviglumis* from the Balsas region except populations Teloloapan (**1**) and Alcholoa (**2**). In the dendogram, the northern Jalisco highland Guachinango (**12**) population is closest to maize. The network also highlights the intermediate position of certain

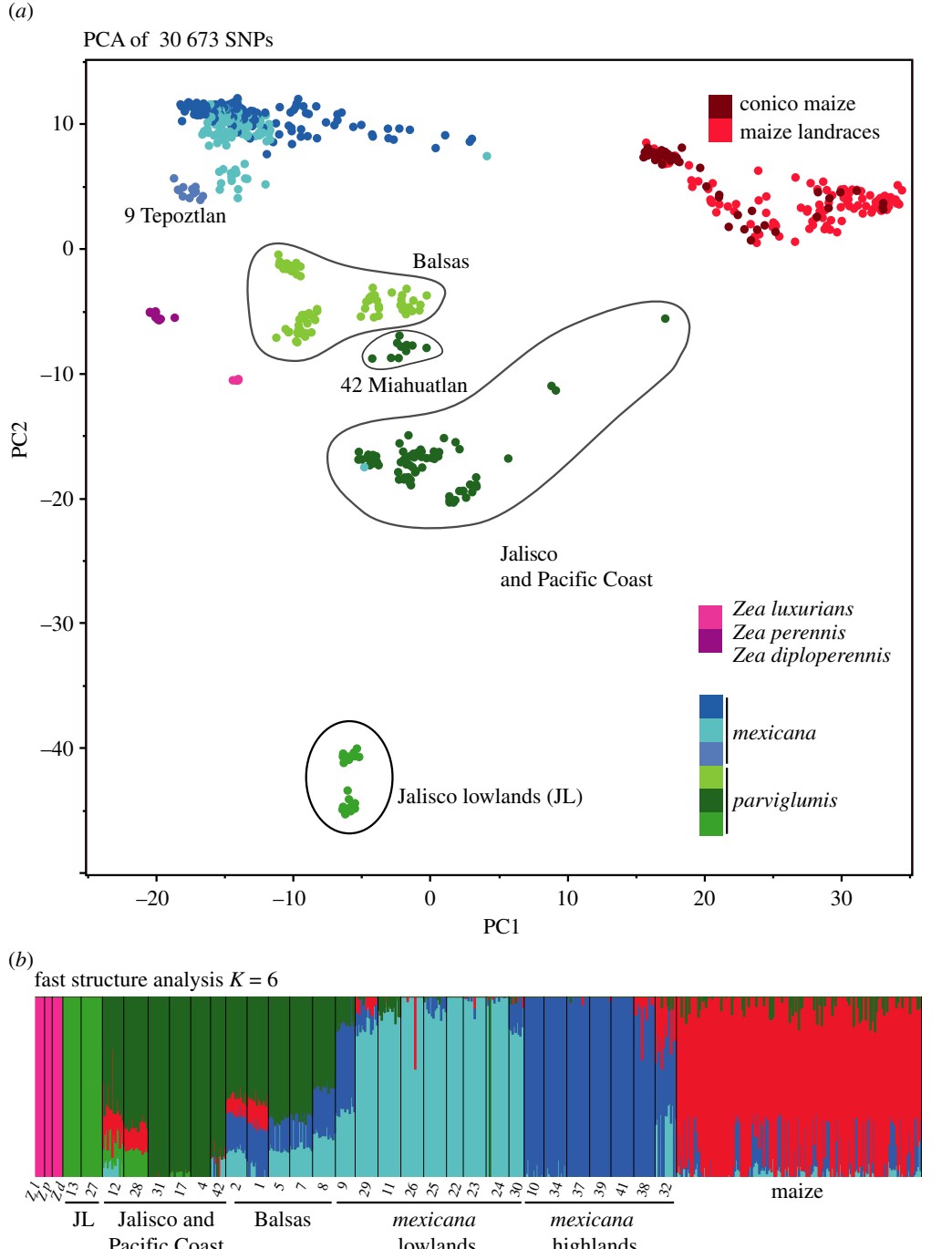

**Figure 3.** (*a*) PCA of 30 673 SNPs. (*b*) fastStructure analysis with the best *K* value. Numbers and colours correspond to those of figure 2.

*parviglumis* populations, in particular Tepoztlan (**9**) from the Balsas region (figure 2; electronic supplementary material, figure S1) which clustered closer to *mexicana* in the PCA analyses (figure 3*a*). These results are very similar to those obtained with GBS data, where maize is clearly divergent from teosinte, and with some Balsas populations more related to *mexicana* in the dendogram (electronic supplementary material, figure S2A), and with a Jalisco *parviglumis* population closely clustering with maize (electronic supplementary material, figure S2).

A second potential explanation for the close relationship between maize and *parviglumis* from Jalisco (and particularly from Guachinango **12**) is introgression. We examined our SNPchip data for potential evidence of introgression among taxa and populations by first examining genetic structure, based on the fastStructure algorithm. For these analyses, *K* = 6 was the optimal number of clusters (figure 3*b*),

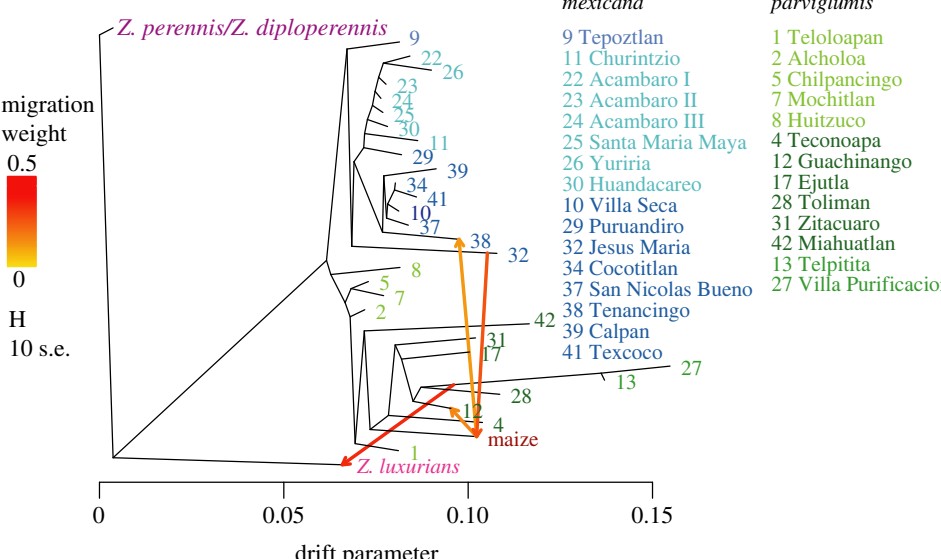

**Figure 4.** Population graph depicting relationships among populations, including migrations events, obtained with TreeMix with 30 673 SNPs. Two migration events were inferred considering shared polymorphism with *Z. perennis* and *Z. diploperennis*. The drift parameter indicates the intensity of genetic drift and branch length is proportional to the amount of genetic drift present. The colour of the migration lines indicates the percentage of loci shared between populations. The covariance matrix used to infer migration events was based on allele frequencies and can be seen in electronic supplementary material, figure S3. Numbers and colours correspond to populations in figure 2.

followed closely by $K = 7$ (electronic supplementary material, figure S3). For $K = 6$, the outgroups (pink) form a distinct group, whereas *mexicana* is split in two groups correlated to altitude (light and dark blue), with a single very admixed population (Tepoztlan **9**) found in a potential contact zone between *parviglumis* and *mexicana*. In *parviglumis*, we detected differentiation into several groups: a homogeneous lowland Jalisco group including two populations (lime green), a group of six populations from Jalisco and the Pacific Coast (dark green), and a very admixed group of five populations from Balsas including genetic components of highland Jalisco and Pacific *parviglumis*, but also of lowland and highland *mexicana*. High admixture between *parviglumis* and *mexicana* can be due to geographical proximity and has been previously reported [24].

For $K = 6$, maize is mostly homogeneous, with some landraces showing more admixture with *mexicana* and *parviglumis*. These landraces are mostly from the conico group found in the highlands of Central Mexico (figure 2), where ancient admixture with teosinte has been reported [20,24]. However, it is noteworthy that Jalisco populations Guachinango (**12**) and Toliman (**28**), which are not geographically close nor environmentally similar to highland maize, show introgression with maize. The second best grouping was $K = 7$, which split maize into two groups, but with no obvious structure (electronic supplementary material, figure S3) [35]. Overall, the same results were obtained with the GBS data, where some individuals from the Balsas region were also strongly admixed with *mexicana* (electronic supplementary material, figure S2B).

To further explore the relevance of introgression, we inferred migration events in a maximum-likelihood framework [46], using *Z. perennis* and *Z. diploperennis* as outgroups to infer ancestral states of alleles and to distinguish gene flow from shared ancestral polymorphism [46]. As no significant genetic structure was detected within our dataset of maize landraces from Arteaga *et al.* [35], we used all 161 of the individuals from the 43 landraces as one population in these analyses. TreeMix [46] produces a population graph that resembles (but is not) a phylogenetic dendogram (figure 4). In a population graph, the branch lengths are proportional to the amount of genetic drift, and the migration events are inferred based on high deviations from a covariance matrix (electronic supplementary material, figure S4, blue and black colours). In the population graph, maize is placed within the Jalisco and Pacific Coast *parviglumis* group (figure 5). Overall, the graph supports maize domestication from *parviglumis* populations from Western Mexico, not the Balsas Basin as previously thought. Our results suggest that *parviglumis* from the Balsas region is highly admixed with *mexicana*, as shown by the structure analysis

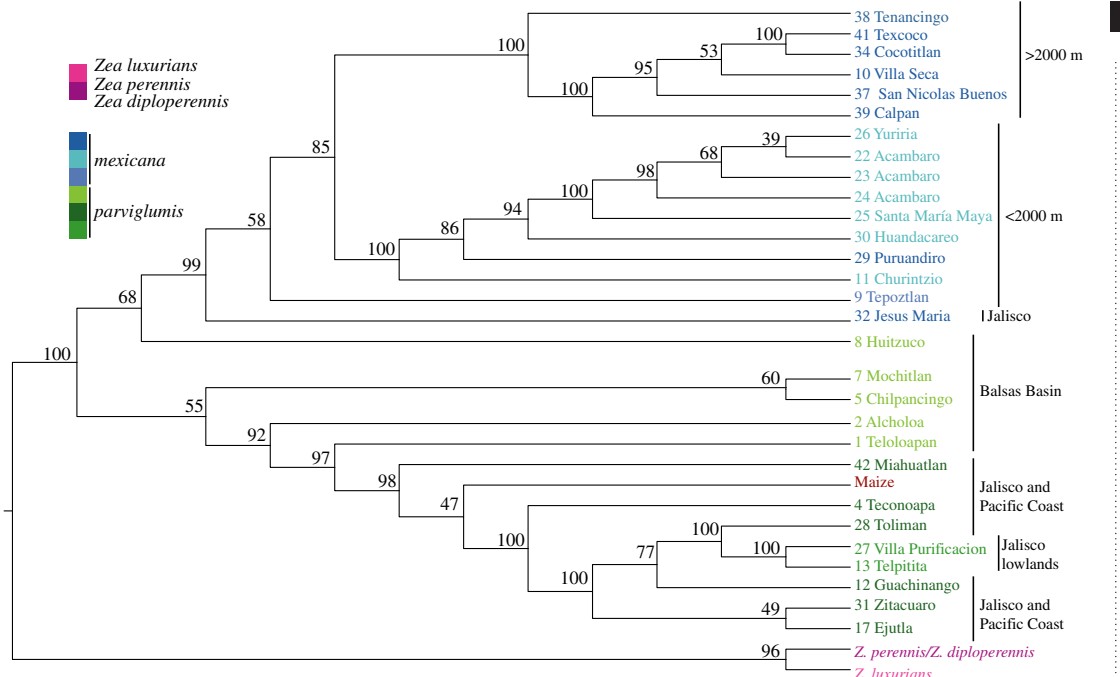

**Figure 5.** Consensus population dendogram obtained from 100 bootstrap replicates of population graphs using Treemix 1.2. Colours and numbers correspond to figures 2 and 3a. *Zea perennis*, *Z. diploperennis* and *Z. luxurians* were used as outgroups. The topology represents the consensus of population relationships based on allelic frequency covariance for each replicate. Bootstrap support is shown as numbers next to each node.

(figure 3b), and by the low support of the relationships of some Balsas populations with either *parviglumis* and *mexicana* (figure 5), and by the population ancestry analysis that groups Balsas populations with *mexicana*, based on admixture proportions, along an altitudinal cline estimated by TESS3 (electronic supplementary material, figure S5). Similar studies show that equal admixture proportions can indicate recent hybridization rather than ancestral polymorphism [49,50].

Once ancestral polymorphism was identified, four main potential admixture events were detected (arrows in figure 4): one from a highland *mexicana* population (Jesus Maria **32**, Jalisco state) to maize and another from maize to another *mexicana* highland population (Tenanaingo **38**, Tlaxcala state), another from lowland Jalisco (Villa Purificacion **27**) to *Z. luxurians*, and one from maize to *parviglumis* from Guachinango (**12**), in Jalisco. The later explains the admixture proportions and genetic similarity of that population with maize and shows that introgression events did not happen homogeneously (figure 4). Introgression between maize and teosinte has been reported, but previous studies could not differentiate between contemporary processes and ancestral introgression [20,24]. Our migration inference with TreeMix identifies migration events in opposite directions: one from maize to a *mexicana* population in Tlaxcala (Tenanacingo **38**), and another from an ancestral *mexicana* from Jalisco (Jesus Maria **32**) to maize. The ancestral nature of the later event is indicated by the position of the migration arrow along the branch, instead of the tip of the modern populations. Such ancestral introgression has also been confirmed in different studies [20,24], and by our own admixture analysis that shows shared ancestry between conico maize landraces and *mexicana* from central Mexico (electronic supplementary material, figure S5). However, gene flow between maize and *parviglumis* was not previously reported [23], suggesting that the sampling design is extremely important to identify these events when populations are so genetically heterogeneous as in *parviglumis*. Altogether, we have presented a novel analysis of genetic diversity and divergence in *Zea mays*, including wild and domesticated samples. Our PCA results show a high degree of genetic structure and differentiation of *parviglumis* populations (figure 3a,b), particularly the Jalisco lowland populations of Telpitita (**13**) and Villa Purificacion (**27**) on the Pacific Coast that form a distinct cluster. This genetic heterogeneity highlights the importance of having a dense enough sampling in the mountains of Jalisco, neighbouring Michoacan state, and the Pacific Coast, which have not been considered in previous domestication studies, despite the area's high biological and cultural diversity [16,20,31].

Some authors have regarded Jalisco as an important centre of domestication in Mesoamerica because the wild relatives of all main crops (maize, squash and beans) are distributed there and at least one of them (beans) was domesticated in the region [31]. Moreover, genetic analyses of *Phaseolus vulgaris* show that the Jalisco accessions are basal to cultivated beans [32,34,51]. Unfortunately, the inherent ascertainment bias of our data prevents us from estimating accurate divergence dates [52]. However, a whole genome genotyping approach together with ancient DNA techniques could help us clarify the question of the time and place of maize domestication.

Altogether, our results reveal an ancient divergence between maize and teosinte and a close relationship with the Jalisco-Pacific Coast populations. These populations (dark green; figures 3–5) may represent 'true' *parviglumis* populations, as opposed to the admixed Balsas populations of Teloloapan (**1**), Alcholoa (**2**) and Chilpancingo (**5**) (light green; figures 3–5; electronic supplementary material, figure S4 and table S3). This pattern suggests that other populations from Western Mexico may be better candidates for maize domestication than the Balsas Basin. The current maize gene pool has an important contribution from certain *mexicana* populations (figure 4), and in turn, maize has contributed to the gene pool of some teosinte populations and subspecies. These events were not widespread and occurred independently, which suggests that the domestication process has included a contribution from some but not all populations of *parviglumis* and has received an important genetic input from some, but not all *mexicana* populations. This does not fit the previous simple model of a single domestication followed by a bottleneck ([6,19] ; figure 1*a*), but a complex scenario with ongoing introgression and contributions from different wild relatives (*mexicana*; figure 1*b,c*). It has been proposed that the introgression with highland *mexicana* populations allowed maize to grow in high-altitude central Mexico [24], which means that at least another round of selection took place after the initial domestication from lowland relatives, similar to a multiple domestication/improvement model with continuous gene flow with wild relatives (figure 1*c*).

## 4. Conclusion

In conclusion, our results showed more introgression events than previously reported, between maize and western populations of *mexicana* and *parviglumis*. These results highlight the importance of having a broad representation of potential wild relatives in order to understand the domestication process of modern crops. In the present example, the remarkable genetic differentiation of *parviglumis* populations means that sampling is crucial to identify the closest relatives to maize, as only the Western and Pacific Coast populations seem to share a gene pool with maize. Given the complexities of the gene flow patterns and the high probability that domestication of maize was a protracted and complex process, we can conclude that pinpointing a single geographic region as centre of origin for maize is a mirage and may never be solved, but we also suggest that the Balsas Basin in not a probable area for it.

A broader sampling strategy of maize wild relatives showed that the domestication process was long and complex, incorporating genetic material from different sources, from Jalisco and Pacific Coast of Mexico. The ongoing gene flow between maize and wild relatives, as well as the continuous improvement of landraces highlights the need to protect their wild relatives from genetic erosion and/or extinction, so they can remain a source of genetic diversity for future generations.

Ethics. No special collecting permits were required for this work.

Data accessibility. All teosinte genotypes are available in the Dryad Digital Repository (https://doi.org/10.5061/dryad.sqv9s4n04 [37]), and maize landrace genotypes are available in http://datadryad.org/resource/doi:10.5061/dryad.4t20n [36].

Authors' contributions. A.M.-L. performed the analyses and wrote the manuscript; L.E.E. designed the study and improved the manuscript; J.A.A.-L. and A.V.-L. performed laboratory work and preliminary bioinformatics analyses; D.P. provided all the maize samples and SNP data. All authors gave final approval for publication.

Competing interests. The authors declare no competing interests.

Funding. This work was supported by grants CB2011/167826 awarded to L.E.E., A.M.-L. and A.V.-L. (CONACYT Investigación Científica Básica), CN-10-393 awarded to L.E.E. (UC MEXUS-CONACYT) and M12-A03 ECOS Nord France -CONACYT-ANUIES grant no. 207571 awarded to L.E.E. Maize landrace sampling was supported by a SEMARNAT-CONABIO grant awarded to D.P.

Acknowledgements. Maize landraces collection and genotyping was funded by Secretaría de Medio Ambiente y Recursos Naturales (SEMARNAT) through a grant to CONABIO awarded to D.P. We are grateful to Erika Aguirre-Planter and Laura Espinosa-Asuar for technical support, and to Jeffrey Ross-Ibarra, Brandon Gaut, Maud Tenaillon and Peter Tiffin for their comments that helped improve this manuscript. This paper was written during a sabbatical leave of L.E.E. in the University of Minnesota, Department of Plant and Microbial Biology in Peter Tiffin's laboratory, with support of the programme PASPA- DGAPA, UNAM.

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
