## [Reviewer comments · Royal Society Open Science]

Review History

RSOS-191545.R0 (Original submission)

Review form: Reviewer 1

Is the manuscript scientifically sound in its present form?

Yes

Are the interpretations and conclusions justified by the results?

Yes

Is the language acceptable?

Yes

Do you have any ethical concerns with this paper?

No

Have you any concerns about statistical analyses in this paper?

No

Recommendation?

Accept with minor revision (please list in comments)

Comments to the Author(s)

The article by Moreno-Letelier is an excellent manuscript but some more care is needed in the presentation. As rendered on my printout Figure 2 and all the Supplementary Material is of too low Quality. Furthermore the Results are probably overly concise and the data should be more fully embraced at Points. The authors should better proofread as a number of careless Errors are present in the manuscript - Alteration of Fonts, incorrect use of numbers and italics etc etc, These minor Errors currently slightly detract from what is in my opinion a thorough study. However, I must admit I feel that I have not the Expertise to Judge the statistics so a critical Question remains as to how These bear up.

Review form: Reviewer 2 (Charles Clement)**Is the manuscript scientifically sound in its present form?**

Yes

Are the interpretations and conclusions justified by the results?

Yes

Is the language acceptable?

Yes

Do you have any ethical concerns with this paper?

No

Have you any concerns about statistical analyses in this paper?

No

Recommendation?

Accept with minor revision (please list in comments)

Comments to the Author(s)

General

This is a well-designed, well-analyzed and well-written study that sheds new light on maize domestication. Recent studies suggested that maize may not have been domesticated in the Balsas River basin, as previously thought based on isoenzymes, SSR, SNPs and archaeological analyses. As the authors show quite clearly, early studies were misled by inappropriate sampling strategies. The current study has a much more appropriate sampling strategy and, not surprisingly, found different results. The authors are careful to point out that even their strategy may not be sufficient, but it is certainly better and they identify ways to improve it further for future studies. I believe that this study will be well cited because it shows two important conclusions: sampling strategies are important; maize was not domesticated in the Balsas River basin. My recommendation is to accept with minor revision.

Specific

Title: The title is not a good summary of the study's conclusions. Yes, gene flow is important, but it is only a part of the results that lead to the main conclusion: maize was not domesticated in the Balsas River basin. This title may be the reason that this manuscript was transferred from Proceedings B to Open Science!

Abstract: This is well-written but could benefit from strengthening the conclusions.

Introduction: This is well-written and appropriate. A little more space might be dedicated to the most recent studies that suggest that the Balsas River basin is not the origin of domestication of maize.

M&M: Good.

R&D: Good.

Conclusions: There is no separate Conclusions section. Rather, the authors use 3 paragraphs at the end of R&D to present their conclusions. The final paragraph takes the focus off of their real conclusions and could be eliminated. Either create a Conclusions section or re-write the final 3 paragraphs, transforming them into one solid paragraph.

References: All are appropriate. Because maize is not my primary area of study, I do not know if something important is missing. Not all references are in Open Science format, which may be because the author's citation manager was not cleaned correctly when citations were imported.

Figures: All are appropriate. Most can be improved. I made suggestions in the annotated manuscript. All could benefit from improved captions, especially those in Supplementary Materials.

Tables: I did not examine these. I assume that the authors provided good passport data for their samples, since they are so careful with their sampling strategy.

Please see annotated manuscript for detailed suggestions, including some assistance with English.

Decision letter (RSOS-191545.R0)

27-Jan-2020

Dear Dr Moreno-Letelier

On behalf of the Editors, I am pleased to inform you that your Manuscript RSOS-191545 entitled "The relevance of gene flow with wild relatives in understanding the domestication process" has been accepted for publication in Royal Society Open Science subject to minor revision in accordance with the referee suggestions. Please find the referees' comments at the end of this email.

The reviewers and handling editors have recommended publication, but also suggest some minor revisions to your manuscript. Therefore, I invite you to respond to the comments and revise your manuscript.

- Ethics statement

- Data accessibility

It is a condition of publication that all supporting data are made available either as supplementary information or preferably in a suitable permanent repository. The data

accessibility section should state where the article's supporting data can be accessed. This section should also include details, where possible of where to access other relevant research materials such as statistical tools, protocols, software etc can be accessed. If the data has been deposited in an external repository this section should list the database, accession number and link to the DOI for all data from the article that has been made publicly available. Data sets that have been deposited in an external repository and have a DOI should also be appropriately cited in the manuscript and included in the reference list.

If you wish to submit your supporting data or code to Dryad (<http://datadryad.org/>), or modify your current submission to dryad, please use the following link:
<http://datadryad.org/submit?journalID=RSOS&manu=RSOS-191545>

- **Competing interests**

- **Authors' contributions**

- **Acknowledgements**

- **Funding statement**

Because the schedule for publication is very tight, it is a condition of publication that you submit the revised version of your manuscript before 05-Feb-2020. Please note that the revision deadline will expire at 00.00am on this date. If you do not think you will be able to meet this date please let me know immediately.

If your manuscript is newly submitted and subsequently accepted for publication, you will be asked to pay the article processing charge, unless you request a waiver and this is approved by Royal Society Publishing. You can find out more about the charges at <https://royalsocietypublishing.org/rsos/charges>. Should you have any queries, please contact openscience@royalsociety.org.

Kind regards,
Andrew Dunn
Royal Society Open Science Editorial Office

on behalf of Dr James Locke (Associate Editor) and Steve Brown (Subject Editor)
openscience@royalsociety.org

Associate Editor Comments to Author (Dr James Locke):

Associate Editor: 1

Comments to the Author:

The reviewers appreciated the interesting and thorough data presented, but have proposed some minor corrections to improve data presentation before the article is suitable for publication.

Reviewer comments to Author:

Reviewer: 1

Comments to the Author(s)

The article by Moreno-Letelier is an excellent manuscript but some more care is needed in the presentation. As rendered on my printout Figure 2 and all the Supplementary Material is of too low Quality. Furthermore the Results are probably overly concise and the data should be more fully embraced at Points. The authors should better proofread as a number of careless Errors are present in the manuscript - Alteration of Fonts, incorrect use of numbers and italics etc etc, These minor Errors currently slightly detract from what is in my opinion a thorough study. However, I must admit I feel that I have not the Expertise to Judge the statistics so a critical Question remains as to how These bear up.

Reviewer: 2

Comments to the Author(s)

General

This is a well-designed, well-analyzed and well-written study that sheds new light on maize domestication. Recent studies suggested that maize may not have been domesticated in the Balsas River basin, as previously thought based on isoenzymes, SSR, SNPs and archaeological analyses. As the authors show quite clearly, early studies were misled by inappropriate sampling strategies. The current study has a much more appropriate sampling strategy and, not surprisingly, found different results. The authors are careful to point out that even their strategy may not be sufficient, but it is certainly better and they identify ways to improve it further for future studies. I believe that this study will be well cited because it shows two important conclusions: sampling strategies are important; maize was not domesticated in the Balsas River basin. My recommendation is to accept with minor revision.

Specific

Title: The title is not a good summary of the study's conclusions. Yes, gene flow is important, but it is only a part of the results that lead to the main conclusion: maize was not domesticated in the Balsas River basin. This title may be the reason that this manuscript was transferred from Proceedings B to Open Science!

Abstract: This is well-written but could benefit from strengthening the conclusions.

Introduction: This is well-written and appropriate. A little more space might be dedicated to the most recent studies that suggest that the Balsas River basis is not the origin of domestication of maize.

M&M: Good.

R&D: Good.

Conclusions: There is no separate Conclusions section. Rather, the authors use 3 paragraphs at the end of R&D to present their conclusions. The final paragraph takes the focus off of their real conclusions and could be eliminated. Either create a Conclusions section or re-write the final 3 paragraphs, transforming them into one solid paragraph.

References: All are appropriate. Because I maize is not my primary area of study, I do not know if something important is missing. Not all references are in Open Science format, which may be because the author's citation manager was not cleaned correctly when citations were imported.

Figures: All are appropriate. Most can be improved. I made suggestions in the annotated manuscript. All could benefit from improved captions, especially those in Supplementary Materials.

Tables: I did not examine these. I assume that the authors provided good passport data for their samples, since they are so careful with their sampling strategy.

Please see annotated manuscript for detailed suggestions, including some assistance with English.

Author's Response to Decision Letter for (RSOS-191545.R0)

See Appendix A.

Decision letter (RSOS-191545.R1)

27-Feb-2020

Dear Dr Moreno-Letelier,

It is a pleasure to accept your manuscript entitled "The relevance of gene flow with wild relatives in understanding the domestication process" in its current form for publication in Royal Society Open Science.

The Editors are pleased to accept your paper, but note that the manuscript has a number of typographical errors that will need correcting during the proofing process - they identify a number of spacing and spelling errors, for instance. Our copy-editors will perform a 'light touch' edit, but please make sure you carefully check the proof when you receive it to ensure that typographical changes are made.

Due to rapid publication and an extremely tight schedule, if comments are not received, your paper may experience a delay in publication. Royal Society Open Science operates under a

continuous publication model. Your article will be published straight into the next open issue and this will be the final version of the paper. As such, it can be cited immediately by other researchers. As the issue version of your paper will be the only version to be published I would advise you to check your proofs thoroughly as changes cannot be made once the paper is published.

on behalf of Dr James Locke (Associate Editor) and Steve Brown (Subject Editor)
openscience@royalsociety.org

Appendix A

Response to reviewers.

All comments marked in the annotated pdf file were resolved. We add a few comments that will help answer the reviewer's questions.

l. 111 The mid-holocene was used because it is the date for which we have bioclimatic layers, thus, projections could be made.

Fig S1.- The network was changed for a rooted IBS tree. Branches where all individuals from the same population clustered together were collapsed.

Fig S5.- The K value is not the same as the fastStructure analysis because this analysis was performed with a smaller sample of populations in the same geographic regions. In the K=7 analysis, two groups correspond to the outgroups and another two groups to maize. The TESS analysis does not include outgroups and only includes accessions from conical landraces which are in geographic proximity to teosinte. Hence, the K number is not as high.

l. 234 In Arteaga et al. (2016) we tried to find any sort of grouping, either by geographical location, altitude or other variables. The only landraces which showed any type of differentiation were from the conico group.

l. 240 The TreeMix output is referred by the authors (Pickrell & Pritchard 2012) as a graph, not a dendrogram, because it allows for reticulation when migration events are included. In the text I refer to the population relationships based on a covariance matrix as a graph (Figs 4 and 5), and the other reconstructions based on a form of genetic distance (1- Identity By State) as a dendrogram, because those do not allow reticulation (Figs. S1 and S2).

Fig.3 The Mexican Pacific Coast biogeographical region includes the mountain ranges parallel to the coast, therefore, localities can have contrasting altitudes ranging from sea-level to over 1000 m above sea level. Therefore, not all Pacific Coast populations are necessarily low-land